# Associations between Temperature and Influenza Activity: A National Time Series Study in China

**DOI:** 10.3390/ijerph182010846

**Published:** 2021-10-15

**Authors:** Can Chen, Xiaobao Zhang, Daixi Jiang, Danying Yan, Zhou Guan, Yuqing Zhou, Xiaoxiao Liu, Chenyang Huang, Cheng Ding, Lei Lan, Xihui Huang, Lanjuan Li, Shigui Yang

**Affiliations:** 1State Key Laboratory for Diagnosis and Treatment of Infectious Diseases, National Clinical Research Center for Infectious Diseases, Collaborative Innovation Center for Diagnosis and Treatment of Infectious Diseases, The First Affiliated Hospital, Zhejiang University School of Medicine, Hangzhou 310058, China; 11918219@zju.edu.cn (C.C.); zhangxiaobao@zju.edu.cn (X.Z.); 12018219@zju.edu.cn (D.J.); yandanying1995@zju.edu.cn (D.Y.); 12018354@zju.edu.cn (Z.G.); 11818353@zju.edu.cn (Y.Z.); drlxx0601@163.com (X.L.); 21818049@zju.edu.cn (C.H.); dingcheng126@126.com (C.D.); leilansky@163.com (L.L.); 2Subject Teaching (English), College of Foreign Languages, Fujian Normal University, Fujian 350117, China; huangxh1115@163.com

**Keywords:** temperature, influenza activity, H1N1 virus, generalized additive models

## Abstract

Previous studies have reported that temperature is the main meteorological factor associated with influenza activity. This study used generalized additive models (GAMs) to explore the relationship between temperature and influenza activity in China. From the national perspective, the average temperature (AT) had an approximately negative linear correlation with the incidence of influenza, as well as a positive rate of influenza H1N1 virus (A/H1N1). Every degree that the monthly AT rose, the influenza cases decreased by 2.49% (95%CI: 1.24%–3.72%). The risk of influenza cases reached a peak at −5.35 °C with RRs of 2.14 (95%CI: 1.38–3.33) and the monthly AT in the range of −5.35 °C to 18.31 °C had significant effects on the incidence of influenza. Every degree that the weekly AT rose, the positive rate of A/H1N1 decreased by 5.28% (95%CI: 0.35%–9.96%). The risk of A/H1N1 reached a peak at −3.14 °C with RRs of 4.88 (95%CI: 1.01–23.75) and the weekly AT in the range of −3.14 °C to 17.25 °C had significant effects on the incidence of influenza. Our study found that AT is negatively associated with influenza activity, especially for A/H1N1. These findings indicate that temperature could be integrated into the current influenza surveillance system to develop early warning systems to better predict and prepare for the risks of influenza.

## 1. Introduction

Seasonal influenza is a common infectious disease of the respiratory tract, which causes 3–5 million infections and 290,000–650,000 deaths worldwide annually [1,2]. Previous studies have reported that the timing of the influenza epidemic peak varies in latitude, and complex interactions have been found between meteorological factors, human activity patterns, and influenza virus activities [3]. Meteorological factors are considered to link with the spatiotemporal distribution of influenza, including incidences, seasonality, and periodicity [4,5]. In temperate areas, influenza activities typically peak during winter, while in tropical and subtropical zones, the peaks might differ greatly. Some subtropical areas span a single peak per year, in winter or spring/summer, whereas some in both summer and winter, or even last for the whole year [6,7,8,9,10]. In temperate, subtropical, and tropical regions, it has been shown that the low temperature is associated with high levels of influenza activity [11]. It has been reported that low daily temperatures of 0–5 °C significantly increase the risk of influenza incidence [12]. Meanwhile, high temperature can reduce influenza viral replication [13]. An animal study suggested that low temperature could enhance the transmission of the influenza virus, and the duration of virus shedding for the infected hosts housed at 5 °C was significantly longer than that at 20 °C [14]. An Israeli study showed that when the minimum temperature rose by approximately 1.2 °C, influenza activity decreased by 22.0–42.9% [4]. In addition, the impact of weather conditions may vary by influenza virus. A Japanese study showed that only a high proportion of influenza A virus cases is observed during low temperatures and there is not a statistical association with the influenza B virus [15].

Research on the relationship between temperature and influenza activity, including incidence of influenza and susceptible influenza subtypes was limited in China. The discovery and understanding of the impact of temperature on influenza infections could help forecast the influenza epidemic through the surveillance of fluctuating temperature, which is favorable to the preliminary preparations by administrations for disease control [16]. Therefore, in this study, we aimed to explore the effects of temperature on influenza activity in China.

## 2. Materials and Methods

### 2.1. Influenza Epidemiological and Virological Data

In the 1950s, the Chinese government established a routine reporting system for selected infectious diseases, and the available data from 31 provinces in China cover approximately 1.3 billion people. This system has been web-based since 2003 [17]. Influenza belongs to the Class C notifiable infectious diseases in China. In this study, we extracted influenza case data from the notifiable infectious disease report database, which is open and available from the data center of China’s public health sciences [18]. We obtained incidence and mortality data for the influenza cases from 2004 to 2017, stratified by date (month and year) and province. Weekly influenza surveillance virological data stratified by region (northern and southern China) and subtypes (e.g., H1, H3, and B) from 2010 to 2019 were collected from China’s National Influenza Center.

### 2.2. Meteorological Data

Monthly and weekly meteorological data from 839 climate monitor stations during 2004–2019, covering China, were downloaded from the China Meteorological Data Sharing Service System (CMDSSS) (www.data.cma.cn, accessed on 17 April 2021). The monthly and weekly meteorological factors for each monitor station, including average temperature (AT; °C), monthly average relative humidity (ARH; %), monthly average wind speed (AWS; cm/s), monthly average cumulative precipitation (ACP; mm), and monthly average air pressure (AAP; hPa), were collected (Table 1). The meteorological and influenza case data and the meteorological and influenza virological data from the same province were combined to obtain influenza case–weather and influenza virological–weather databases, respectively. These influenza case–weather and influenza virological–weather databases for China were further divided into northern and southern China (Figure 1B).

### 2.3. Generalized Additive Models (GAMs) 

Then, the influenza case–weather and influenza virological–weather databases were classified into several subgroups, including region (northern and Southern China), age (≤14 years, 15–59 years, and ≥60 years). Spearman correlation analysis was conducted, and only one of the highly-correlated variables (*r* > 0.6) was selected to avoid multicollinearity among meteorological factors (Appendix A). For the influenza case–weather database, we developed two GAMs, and for the influenza virological database, we developed three GAMs. We used these GAMs with a negative binomial distribution to initially fit the exposure–response between each meteorological factor and the incidence of influenza. Relative risk (RR) with its 95% confidence interval (CI) was calculated to evaluate the risk of influenza with changes in meteorological factors. An almost linear relationship was found between meteorological factors and the incidence of influenza. Then, we removed the spline function and used the following GAMs:

For the influenza case–weather database:Model1: Log[[E(Yt)]] = β0 + β1(ATt) + β2(ARHt) + β3(MOYt) + s(time,df)+ et; 
Model2: Log[[E(Yt)]] = β0 + β1(ATt) + β2(AWSt) + β3(MOYt) + s(time,df)+ et;

For the influenza virological–weather database:Model1: Log[[E(Yt)]] = β0 + β1(ATt) + β2(ARHt) + β3(AWSt) + β4(MOYt) + s(time,df)+ et;
Model2: Log[[E(Yt)]] = β0 + β1(ATt) + β2(ARHt) + β3(MOYt) + s(time,df)+ et; 
Model3: Log[[E(Yt)]] = β0 + β1(ATt) + β2(AWSt) + β3(MOYt) + s(time,df)+ et;
where Yt denotes the month’s number of influenza cases and weekly positive of influenza virus at time t, respectively; β0 is the intercept; β(AT), β(ARH), β(AWS), and β(MOY) denote the corresponding regression coefficients of AT, ARH, AWS, and MOY (month of year), respectively; MOY is the month ordinal used to control seasonal trends; s (time) is the penalized spline to control the long-term trend; and et is the error term. Then, we calculated the excess risk (ER%, (exp(β) − 1) × 100%) to indicate the effects of meteorological factors on the incidence of influenza. All analyses were performed by R software (R Core Team, Vienna, Austria) with the mgcv package for GAMs.

## 3. Results

### 3.1. The Incidences of Influenza in China 

A total of 1,986,536 influenza cases were reported in China from 1 January 2004 to 31 December 2017, with an average annual incidence 10.41/100,000 (Figure 1A). The incidence of influenza in the ≤14 years age group (32.72/100,000) was higher than those in the 15–59 years (5.82/100,000) and ≥60 years (5.80/100,000) groups (Figure 1C). The rate in southern China (11.22/100,000) was higher than that in northern China (9.43/100,000) (Figure 1D).

### 3.2. The Exposure–Response and Excess Risk of Monthly AT to the Incidence of Influenza

The GAMs found that there was an approximately negative linear correlation between AT and influenza incidences in China with an ER of −2.49% (95%CI: −3.72% to −1.24%). An AT in the range of −5.35 to 18.31 °C had a significant effect on the incidence of influenza, and the risk of influenza cases reached a peak at −5.35 °C (AT) with RRs of 2.14 (95%CI: 1.38–3.33). After stratification analysis, the temperature was negatively associated with the incidence of influenza in all age groups. The contributions of temperature for influenza activity were significant in northern China, with an ER of −4.54% (95%CI: −3.72% to −1.24%). An AT in the range of −12.22 to 19.39 °C had a significant effect on the incidence of influenza, and the risk incidence of influenza cases reached a peak at −12.22 °C with RRs of 5.22 (95%CI: 2.87–9.49) in northern China (Figure 2).

### 3.3. The Activity of Influenza Virus in China

From the first week of 2010 to the last week of 2019, a total of 2,070,967 samples were tested. The positive rate of total influenza virus including A/H1N1, A/H3N2, A/unsubtyped, and influenza B virus (B/Victoria lineage, B/Yamagata lineage and B/unsubtyped) reached 15.01%.The influenza A H3N2 virus (5.30%) was the dominant subtype in mainland China. In northern China, 761,627 samples were tested, with a 15.10% positive rate of total influenza virus; meanwhile, in southern China, 1,309,340 were tested, with a 14.96% positive rate of total influenza virus (Figure 3).

### 3.4. The Exposure–Response and Excess Risk of Weekly AT to the Positive Rate of Influenza Virus

From the national perspective, weekly AT had an approximately negative linear correlation with the positive rate of A/H1N1, with an ER of −5.28% (95%CI: −9.96% to −0.35%). A weekly AT in the range of −3.14 to 17.25 °C had a significant effect on the positive rate of H1N1. The risk of A/H1N1 reached a peak at −3.14 °C (AT) with RRs of 4.88 (95%CI: 1.01–23.75) (Appendix A). After region stratification analysis, both the positive rate of A/H1N1 and the influenza B virus were significantly negatively associated with weekly AT in northern China; however, there was no significant finding in southern China (Figure 4).

## 4. Discussion

In 2017, influenza was ranked as a third-class C notifiable disease in China [19]. It poses a great threat to the public health of all age groups, especially in the ≤14-year-old group. The influenza H3N2 virus is the dominant subtype in China. An approximately negative linear correlation between temperature and influenza activity was found in China, and this association was more significant in northern China and for the influenza H1N1 virus.

AT was negatively associated with the incidence of influenza in our study. This finding is consistent with previous studies. Liu indicated that a 5 °C decrease in the minimum temperature causes an increase of 8% in influenza cases after a one-week lag [20]. It was also previously observed that a 1 °C decrease in temperature increases the estimated risk of influenza occurrence by 11% among military conscripts in Finland [21]. The spread of influenza virus is more stable and efficient under colder conditions. At the same time, people may reduce their outdoor activities and gather in crowded indoor environments with closed windows and doors on cold days, which may increase the risk of influenza virus transmission [8].

In a previous study, the associations between temperature and different influenza viruses were not conclusive. In Germany, the negative association of temperature with Flu-A hospitalization was found, but not with Flu-B [22]. In Hong Kong (a city in southern China), the occurrence of Flu-B decreased when temperature increased; however, there was no significant finding for Flu-A [23]. In our study, AT was negatively associated with the activity of the influenza H1N1 virus in China. In Uganda, compared to influenza H3N2 and B virus, a lower temperature was also marginally significantly associated with higher H1N1 activity [15]. Flu-A virus seems to change its antigen more frequent than Flu-B virus [24], which may influence the sensitivity to climate variability. In a study investigating the association between environment and social conditions and H1N1 mutation, the author found that temperature was the main factor associated with the mutation of A/H1N1 viruses [25]. However, there is still limited information on the interaction of temperature, evolution of the influenza virus, and human activity on influenza transmission. Further research is needed to explore the roles of atmospheric factors, human behavior, and pathogeny of the influenza virus on influenza transmission and to evaluate the contributions of each part.

In northern China, weekly AT was negatively associated with the activity of B virus. This result is similar to a study conducted in a northern Chinese city, Shanghai. Both influenza A and B viruses had negative linear correlations with temperature and both the risk of Flu-A and Flu-B reached a peak at 1.4 °C [26].

Compared to southern China, the effects of temperature on influenza activities were more significant in northern China. In southern China, the seasonality of influenza was also complex. Southern China has a subtropical climate and is warmer than northern China. In winter, in northern China, the typical moderate climate with lower temperature and higher air pressure facilitate the survival and spread of the virus. Furthermore, the temperature in northern China usually drops faster than in southern China. Under these conditions, people could be more susceptible to influenza infection because they do not have enough time to adapt to the sudden fall in temperature [27].

Additionally, high AWS was found to increase the risk of influenza case in this study. This finding was also reported by Jean-Baptist du Prel [22]. A field study in horses indicated that there is an increased impact of influenza as the wind speed increases to 30 km/h [28]. This observation may support the hypothesis that high wind speed could accelerate air circulation, thus resulting in longer travel of airborne aerosols and increasing the risk of influenza virus transmission [29,30,31]. Another possible explanation is that strong wind could increase body susceptibility to influenza through lowering the body temperature, along with vasoconstriction in the respiratory tract mucosa. Innate immune responses would be inhibited in this case.

However, our study has several limitations. First, data on human behaviors in response to different weather conditions were not available for this study. We could not assess the role of human behavior changes in different weather conditions on influenza transmission. Further research is needed to include data on meteorological factors, human behaviors, and influenza to explore the potential relationship between them. Second, the data collected and analyses performed were from a national perspective, which might not capture the heterogeneity at the regional level. We therefore divided the data into southern China and northern China to account for regional differences.

## 5. Conclusions

The average annual incidence of influenza was 10.41/100,000 in China and was high in the ≤14-year-old age group. The influenza H3N2 virus was shown to be the dominant subtype in China. AT was approximately negatively linearly associated with the incidence of influenza, as well as A/H1N1 activity from the domestic perspective. Monthly AT in the range of −5.35 to 18.31 °C had significant effects on the incidence of influenza. Weekly AT in the range of −3.14 to 17.25 °C had significant effects on the positive rate of A/H1N1. These findings indicate that temperature could be integrated into the current influenza surveillance system to develop early warning systems to better predict and prepare for the risks of influenza.

## Figures and Tables

**Figure 1 ijerph-18-10846-f001:**
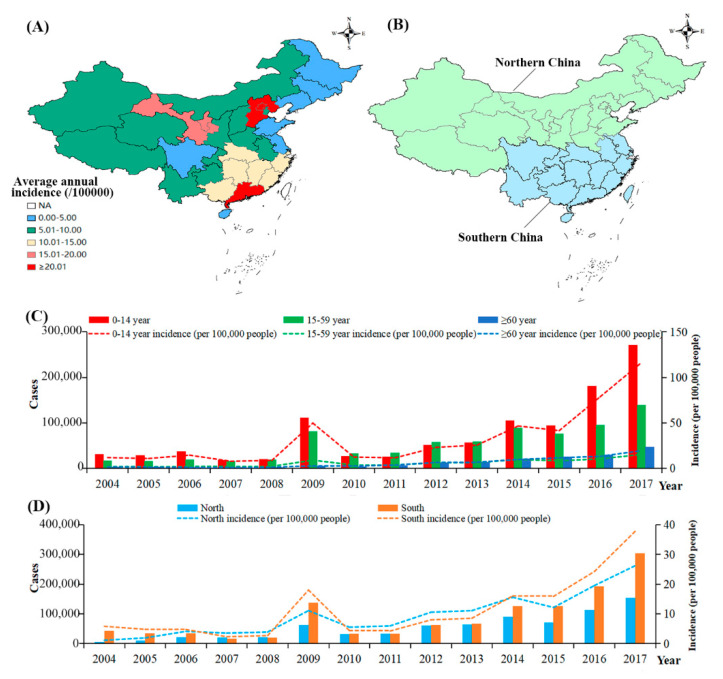
The incidence of influenza in China. (**A**). Geographical distribution of influenza in China; (**B**). Geographical distribution of Southern and Northern China; (**C**). The incidence of influenza in three age group; (**D**). The incidence of influenza in northern and southern China.

**Figure 2 ijerph-18-10846-f002:**
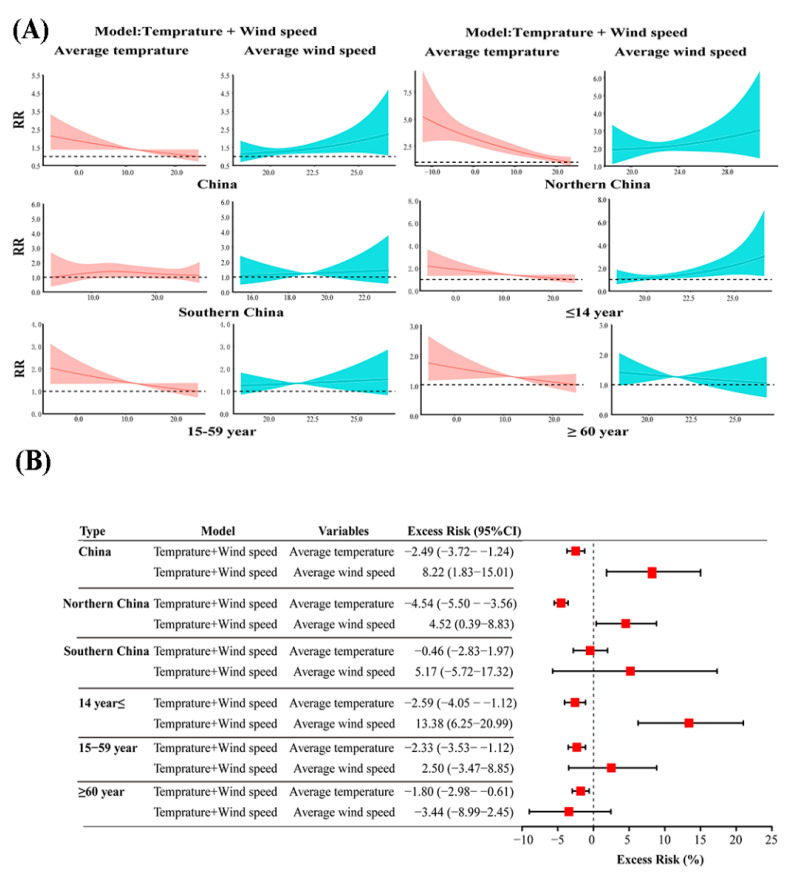
The exposure–response and excess risk of monthly AT on the incidence of influenza. (**A**). The exposure–response of monthly AT on the incidence of influenza; (**B**). The excess risk of monthly AT on the incidence of influenza.

**Figure 3 ijerph-18-10846-f003:**
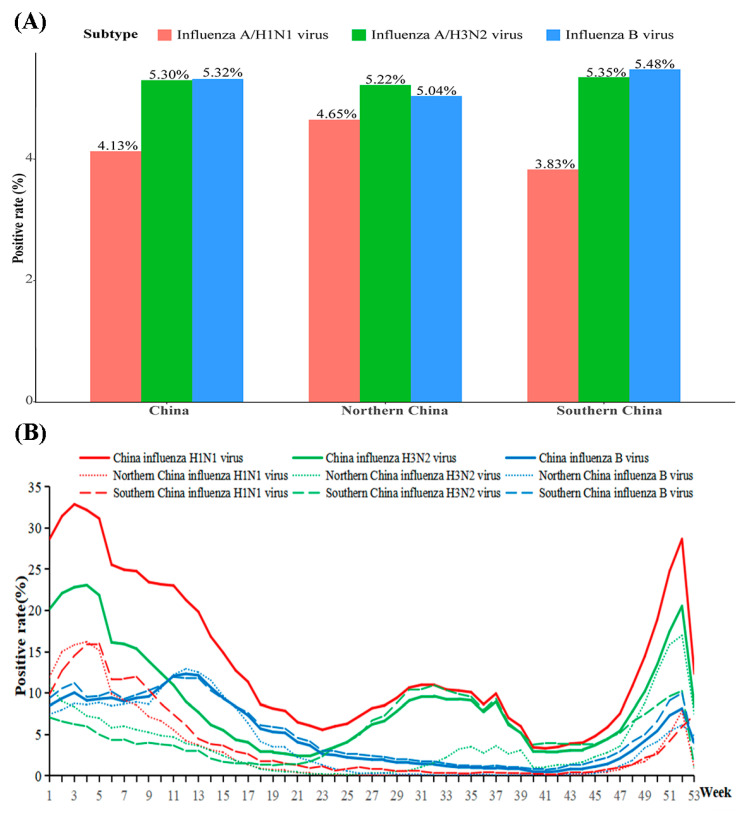
The activity of influenza virus in China. (**A**) The positive rate of influenza virus in China; (**B**) The activity of influenza virus in China by week.

**Figure 4 ijerph-18-10846-f004:**
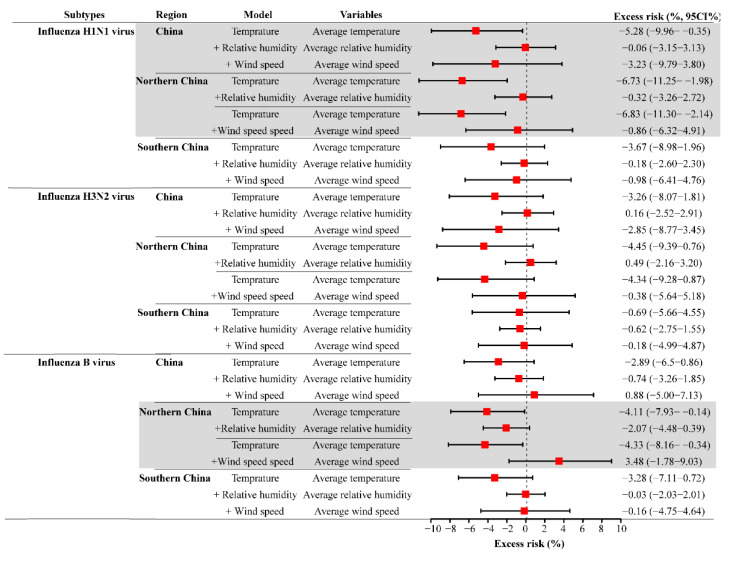
The exposure–response and excess risk of weekly AT to the positive rate of influenza virus.

**Table 1 ijerph-18-10846-t001:** Description of the monthly/weekly influenza and meteorological data.

**Monthly Influenza Cases and Meteorological Data**								
**Regions**	**Factors**	**Data Source**	**Number of Months**	**Min.**	**P (25th)**	**Median**	**P (75th)**	**Max.**
China	Influenza cases	Notifiable infectious disease report database	168	1179	3567	7210.5	13,953	135,398
	AT, °C	CMDSSS	168	−5.35	3.82	13.19	20.12	24.77
	ARH, %	CMDSSS	168	53.57	59.86	64.64	69.92	74.22
	AWS, cm/s	CMDSSS	168	18.3	19.68	20.9	23.05	26.7
	ACP, mm	CMDSSS	168	6.22	28.4	58.32	113.63	168.44
	AAP, hPa	CMDSSS	168	911.26	915.25	920.82	924.84	936.15
Northern China	Influenza cases	Notifiable infectious disease report database	168	153	1208	3106	5374	58,371
	AT, °C	CMDSSS	168	−12.22	−2.02	9.3	17.43	23.46
	ARH, %	CMDSSS	168	43.44	51.82	58.03	64.54	71.08
	AWS, cm/s	CMDSSS	168	18.43	20.97	22.55	25.2	30.82
	ACP, mm	CMDSSS	168	1.89	8.72	26.54	62.39	173.42
	AAP, hPa	CMDSSS	168	892	896.92	902.06	906.21	924.91
Southern China	Influenza cases	Notifiable infectious disease report database	168	551	1936	3689	8360	77,027
	AT, °C	CMDSSS	168	3.67	11.45	18.25	23.56	26.85
	ARH, %	CMDSSS	168	64.39	71	73.89	76.61	81.54
	AWS, cm/s	CMDSSS	168	15.42	17.97	18.88	20.07	23.37
	ACP, mm	CMDSSS	168	11.28	51.61	98.11	170.13	278.8
	AAP, hPa	CMDSSS	168	935.58	939.64	946.03	949.67	954.66
**Weekly Influenza Virological and Meteorological Data**								
**Regions**	**Factors**	**Data Source**	**Number of Weeks**	**Min.**	**P (25th)**	**Median**	**P (75th)**	**Max.**
China	Influenza H1N1 virus (%)	China National Influenza Center	522	0	0.09	0.49	2.63	35.29
	Influenza H3N2 virus (%)	China National Influenza Center	522	0	0.6	2.44	6.88	32.67
	Influenza B virus (%)	China National Influenza Center	522	0	0.59	1.71	6.35	38.91
	AT, °C	CMDSSS	522	−5.62	3.62	13.74	21.17	26.08
	ARH, %	CMDSSS	522	49.39	61.48	66.1	70.68	77.93
	AWS, cm/s	CMDSSS	522	16.29	19.76	21.19	23.13	29.46
	ACP, mm	CMDSSS	522	0.24	6.79	15.63	26.87	55.65
	AAP, hPa	CMDSSS	522	913.24	919.86	925.39	929.2	941.14
Northern China	Influenza H1N1 virus (%)	China National Influenza Center	522	0	0	0.15	2.03	37.68
	Influenza H3N2 virus (%)	China National Influenza Center	522	0	0.15	1.17	4.47	38.48
	Influenza B virus (%)	China National Influenza Center	522	0	0.09	0.45	3.75	41.16
	AT, °C	CMDSSS	522	−13.65	−2.87	9.61	18.46	24.52
	ARH, %	CMDSSS	522	36.93	51.03	57.34	64.15	74.61
	AWS, cm/s	CMDSSS	522	16.32	21.01	23.04	25.76	34.76
	ACP, mm	CMDSSS	522	0.09	1.77	5.52	14.27	50.26
	AAP, hPa	CMDSSS	522	884.55	892.72	898.73	902.09	917.93
Southern China	Influenza H1N1 virus (%)	China National Influenza Center	522	0	0.09	0.5	2.73	40.53
	Influenza H3N2 virus (%)	China National Influenza Center	522	0	0.59	2.07	6.96	34.16
	Influenza B virus (%)	China National Influenza Center	522	0	0.73	2.08	7.13	39.7
	AT, °C	CMDSSS	522	2.65	11.17	18.68	24.16	28.6
	ARH, %	CMDSSS	522	57.38	72.78	76.12	79.16	85.9
	AWS, cm/s	CMDSSS	522	14.17	17.71	18.93	20.39	26.38
	ACP, mm	CMDSSS	522	0.03	11.25	24.41	39.4	94.63
	AAP, hPa	CMDSSS	522	942.17	949.93	955.91	960.46	968.93

## Data Availability

The raw data supporting the conclusions of this article will be made available by the corresponding author, without undue reservation.

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
