# Peer review of "Associations between Temperature and Influenza Activity: A National Time Series Study in China"

_ijerph, 2021, doi:10.3390/ijerph182010846_

Round 1
Reviewer 1 Report
- Need major English review.
- Table 1: I recomend the authors indicate from which data source (disease report database or meteorological data) those statistical metrics came from. Also, I recomend you also include the effective sample size considered for each statistics metrics. It is usual in any data set to present "NA" values. Because of this, please, include the sample size of each metrics.
- GAM: Why did you chose spearman correlation? What are the variables you considered to proceed with such correlation? The spearman correlation is applied for ordinal qualitative variables or non-parametric quantitative data. However, I saw that many variables are quantitative. So, please, I suggest you clarify what are the pair of ordinal qualitative variables or non-parametric quantitative data you considered for spearman correlation. Otherwise, I suggest you review the use of this approach.
- GAM: The variable MOY is limited between 1 to 12? If so, would the model assume that the same value of Beta-3 is considered to model from 2004 to 2017? For example, can you assume that there was no yearly changes in the environmental scenery? Why did you not include an year variable input (that is, a specific variable for "year")?
Author Response
Dear reviewer:
On behalf of my co-authors, we thank you very much for giving us the opportunity to revise our manuscript. We are very grateful to you for your positive and constructive comments and suggestions regarding our manuscript entitled “Associations of temperature and influenza activity: A national time-series study in China” (Manuscript ID: ijerph-1388526). We have carefully considered your comments and have revised our manuscript according to these comments. We have tried our best to improve the manuscript and have made some changes, which are marked red in the revised manuscript. In addition, to further improve its clarity and readability, the manuscript has been edited for proper English language usage by the highly qualified native English-speaking editors at MDPI. These changes have not influenced the content or framework of the paper. Please find the attached revised version of the manuscript, which we would like to submit for your kind consideration. We sincerely appreciate for your work, and we hope that the corrections will meet with your approval. Once again, thank you very much for your comments and suggestions.
Yours sincerely
Shigui Yang
State Key Laboratory for Diagnosis and Treatment of Infectious Diseases, National Clinical Research Center for Infectious Diseases, Collaborative Innovation Center for Diagnosis and Treatment of Infectious Diseases, The First Affiliated Hospital,Zhejiang University School of Medicine. Email: yangshigui@zju.edu.cn; Telephone: +86 13605705640; Address: 79 Qingchun Road, Hangzhou, China

Reviewer 2 Report
Authors propose generalized additive models (GAMs) to explore the relationship between temperature and influenza activity in China. The main finding is the average temperature was negatively associated with influenza activity, especially for A/H1N1.
The paper is well presented, clear, objective and well written with good figures to present to the reader. Many characteristics of the seasonal impact of influenza are discussed throughout the paper.
From my point of view, the paper is ready to be accepted. I would only improve the quality of Figure 3.
Author Response

(The authors gave the same response as above.)

Reviewer 3 Report
Supplementary material:
Mislabelled figure: Supplementary Fig. 2, Northern China (middle columns), 3rd set of graphs from top: (Inf H3N2) mislabelled
Manuscript:
Questions (for the authors): what is the difference between “China” and “Northern China” and “Southern China”? Is the data for China just the overall compilation of Northern and Southern added together? Please clarify.
While the focus is on atmospheric conditions in this study, what role do human behavior changes in response to those conditions (for example, gatherings held indoors versus outside, more time spent in cars/trains/crowded places, children in schoolrooms all day versus with time outside for play) play in influenza transmission? This is mentioned, but could the authors possibly include (here or in future publications) any behavioral data?
It is good that mutations are discussed as a factor relevant to atmospheric conditions, but to what degree is this a viral endogenous attribute, and to what degree an effect of human crowding indoors (increased transmission and selection for mutated strains)?
The conclusions seem reasonable, and providing additional material to assist in predictive modelling of influenza spread based on weather conditions is a worthy goal.
Author Response

(The authors gave the same response as above.)
